# Validation and Reliability Testing of the Child Oral Impacts on Daily Performances (C-OIDP): Cross-Cultural Adaptation and Psychometric Properties in Pakistani School-Going Children

**DOI:** 10.3390/children9050631

**Published:** 2022-04-28

**Authors:** Farooq Ahmad Chaudhary, Azhar Iqbal, Muhammad Danial Khalid, Nouman Noor, Jamaluddin Syed, Muhammad Nadeem Baig, Osama Khattak, Shahab Ud Din

**Affiliations:** 1School of Dentistry (SOD), Federal Medical Teaching Institution (FMTI)/PIMS, Shaheed Zulfiqar Ali Bhutto Medical University (SZABMU), Islamabad 44000, Pakistan; nur.nouman@gmail.com (N.N.); drshahab728@hotmail.com (S.U.D.); 2Department of Operative Dentistry and Endodontics, College of Dentistry, Jouf University, Sakaka 72388, Saudi Arabia; dr.azhar.iqbal@jodent.org; 3Dental College, HITEC Institute of Medical Sciences, Taxila 47080, Pakistan; drdanial12@gmail.com; 4Oral Basic and Clinical Sciences Department, Faculty of Dentistry, King Abdulaziz University, Jeddah 21589, Saudi Arabia; drjamalsyed@gmail.com; 5Department of Regenerative Dentistry, University of Bari, 70121 Bari, Italy; 6Department of Preventive Dentistry, Jouf University, Sakaka 72388, Saudi Arabia; drmnb22@gmail.com

**Keywords:** C-OIDP, oral health-related quality of life, reliability, Urdu, validity

## Abstract

Background: This study aimed to develop an Urdu version of Child-Oral Impact on Daily Performance (C-OIDP) and assess its reliability and validity for children’s oral health-related quality of life (OHRQoL) assessment in Pakistan. Methods: A total of 200 school-going children aged 11–14 were recruited from two public schools. For the adaptation process, the original English version of C-OIDP was translated into Urdu, reviewed by an expert committee, back-translated into Urdu, and then reviewed again by the same expert committee and pilot tested on 10 children. A clinical examination was carried out to record dental and gingival status followed by a face-to-face interview to measure oral health-related quality of life in children using C-OIDP-U. Reliability, internal consistency, construct and discriminant validity were assessed. Results: The Cronbach’s alpha for C-OIDP-U was 0.69, the mean C-OIDP-U score was 10.2 ± 8.1 and 77.3% of the children reported at least one oral impact. Eating (40.3%) and difficulty in cleaning mouth (38.7%) were the two most impacted daily performances. For construct validity, the associations were significant between the C-OIDP-U score and all subjective oral health measures (*p* < 0.001). For discriminant validity, a significant association was observed between the C-OIDP-U score and clinical oral variables, children with DMFT + dmft ≥ 1, Gingival index > 1 and having malocclusion reported a higher C-OIDP-U score when compared to their counterparts. Conclusion: This study showed that C-OIDP is a valid, reliable and efficient instrument of OHRQoL for use in Pakistani children.

## 1. Introduction

Health is a multidimensional phenomenon that is determined by physical, mental emotional, and social health aspects [1]. Methods traditionally measuring health rely exclusively on clinical indicators and emphasis on disease presence or absence. However, with the emergence of the quality-of-life concept, various health-related quality of life (HRQoL) questionnaires are being utilized to assess the mental and social dimensions of health, which are considered as important in the clinical aspect of health [2]. HRQoL is an individual’s or a group’s perceived physical and mental health, and it assesses the effects of a disease, disability or disorder on their life and wellbeing over time [3]. As oral health is closely related to general health through affecting their oral functions, various measures for oral health-related quality of life (OHRQOL) have been tested to evaluate the level of impact oral health conditions have on an individual’s behavior, self-esteem and social functioning [4].

There are various measures of OHRQOL developed in the past two decades such as Oral Impact on Daily Performance, Oral Health Impact Profile and the Geriatric Oral Health Assessment Index; however, most of these measures were developed for use in adults [5,6,7]. Very few measures were developed specifically for the use in children due to many conceptual and methodological difficulties, such as children’s perception of illness and health that is influenced by age, emotions, language, cognitive development and changes in dental and facial features.

Child-Oral Impact on Daily Performance (C-OIDP) is one of the measures that is developed specifically for children in Thailand [8]. Since its inception, it has been used widely and found to be a valid and reliable measure to assess the impacts of oral condition on daily life among children in various countries [9,10,11,12]. Hence, using this measure in different countries implies that it went through meticulous translation, cross-cultural adaptation in their respective languages and cultures for testing its psychometric properties. Despite C-OIDP being widely used, it has never been used for Pakistani children and no other OHRQOL measure for children is available in the Urdu language. Therefore, this study aimed to develop an Urdu version of Child-Oral Impact on Daily Performance (C-OIDP) and assess its reliability and validity for children’s OHRQoL assessment in Pakistani 11–14-year-old children.

## 2. Materials and Methods

This cross-sectional retrospective study was conducted between October and December 2021 on an opportunity sample selected from school-going children attending government schools in Islamabad, Pakistan. Two government schools were selected that were located near the dental hospital where the main researcher (F.A.C.) worked. The approval of the research was taken from the Ethical review board of the School of dentistry, Shaheed Zulfiqar Ali Bhutto Medical University, Islamabad (Ref. No. SOD/ERB/2022/120). Written informed consent was obtained from the parents and guardians of all the children who participated in this study, after explaining and informing them about the objective of the study. Furthermore, written approval was obtained from the principal of both the included schools to conduct the study.

A sample size of 100–200 is recommended in the literature for the psychometric and validation of questionnaires studies [13]. Therefore, 197 children aged 11–14-year-old were recruited for this study.

### 2.1. Translation

The published steps and guidelines (translation, committee review, back-translation and second committee review) were followed for the translation and validation of the OIDP in the Urdu language [14,15,16]. The translations of the original English C-OIDP were carried out by two bilingual experts (Urdu and English language). The Urdu translated version of C-OIDP (C-OIDP-U) was reviewed for appropriate wording, clarity, interpretation and cultural acceptability by the five-member expert committee. After the review by the expert committee, C-OIPD-U was back-translated into the English language by two different bilingual experts who had no prior knowledge of the original C-OIPD. The same five-member expert committee again reviews the back-translated version for the equivalence, changes and similarities between the original English and back-translated version. Afterward, the final C-OIPD-U was pilot tested on 10 children from a different school to validate its items’ clarity and cultural context. All participants of the pilot test confirmed that they clearly understood the content of the questionnaire.

### 2.2. Questionnaire and Clinical Oral Examination

The C-OIDP-U version of the questionnaire was used in this study to measure the oral health-related quality of life in children. The C-OIDP index was specifically developed by Gherunpong et al. (2004) for assessing children’s oral impacts on daily performances with focusing on 8 life activities such as eating, speaking, cleaning the teeth, sleeping, emotional state, smiling, studying and social contacts [8]. The frequency and severity of each impact were scored on Likert scales ranging from 0 to 3. The impact score for each performance was obtained by multiplying severity and frequency scores (range 0 to 9), and the total impact score was calculated by adding the impact scores of all eight performances (range 0 to 72). This overall score was divided by 72 and multiplied by 100 to obtain the total percentage score.

The data using C-OIPD-U were taken from the children by the main investigator (F.A.C.) by conducting face-to-face interviews. At first, the children were asked about any oral health problem they faced in the last 6 months; if the response was no, then ‘no’ further questions were asked, and the score was considered as 0. However, if they say ‘yes’, then they were further interviewed for severity and frequency for each OIDP daily performance.

### 2.3. Demographic, Behavioural and Subjective Variables

For demographic age and gender, information was recorded during the interview. Furthermore, the interview contained questions regarding the self-perception of oral and general health, as a single item question with three possible answers for each (good, fair and poor), perceived treatment needs (yes, no and do not know) and dental appearance (good, average and bad).

### 2.4. Clinical Examination

Oral examination was carried out by one trained investigator (F.A.C.) with the help of a dental assistant who acted as a recorder to evaluate their oral health condition by following the previously described methodology [17,18,19]. The decayed, missing and filled teeth index (DMFT + dmft) was recorded in both the primary and permanent dentition. The scores were categorized into two groups: one caries-free (DMFT + dmft = 0) and the having caries group (DMFT + dmft ≥ 1) [12]. The gum status of children was examined using the Gingival index (GI), and the scores were categorized into those having healthy gingiva with firm and pink gingiva (GI ≤ 1) and those having unhealthy gingiva with redness, swelling and bleeding on probing (GI > 1) [12]. The occlusion of the children was also examined, and each participant was assigned as either having normal occlusion or having malocclusion based on Angle’s Class I, II and III malocclusions [12]. For the purpose of test–retest reliability, 20 children representing 10% of the sample were reinterviewed after 2 weeks.

### 2.5. Data Analysis

Cronbach’s alpha, item-total and Pearson correlation coefficients were used to assess the internal consistency of C-OIDP-U, and for test–retest reliability, weighted kappa coefficient was used. For construct validity, the children’s C-OIDP-U score was compared with self-perceived oral and general health, perceived treatment needs and dental appearance using ANOVA and independent *t*-test. For discriminant validity, the children’s C-OIDP-U score was compared with clinically examined dental status, gingival health and malocclusion. SPSS version 24 and 0.05 level of statistical significance were used for all data analyses.

## 3. Results

Table 1 represents the demographic characteristics and distribution of impact on daily performances; in total, 197 participants were recruited, and among them, a majority (32.9%) belongs to the 13-year age group and were boys (58.3%). Overall, 77.3% of the participants reported at least one oral impact in the last 3 months. The most prevalent impact on daily performance was reported for eating (40.3%) and cleaning (38.7%), and the lowest impact was reported for studying (9.1%) and contact with people (10.2%). The mean C-OIDP-U score was 10.2 (±8.1) with the highest score for eating (23.3 ± 26.3) and cleaning (15.6 ± 21.5).

The C-OIDP-U showed satisfactory reliability with Cronbach’s alpha of 0.69, and Cronbach’s alpha value decreased when any item was deleted, and no negative correlation was observed. All items were above 0.30 for item-total correlation, and the weighted kappa value was 0.81 for test–retest reliability (Table 2).

The construct validity and the associations between the C-OIDP-U and subjective oral health measures were tested. Children who reported self-rated oral health to be good and fair had significantly lower C-OIDP-U scores than those who reported it to be poor (*p* < 0.001). Similarly, those participants who perceived their general health to be good and fair had significantly lower C-OIDP scores than those who perceived it to be poor (*p* < 0.01). Likewise, children who perceived no need for dental treatment had lower C-OIDP-U scores than those who felt the need or did not know about the need (*p* < 0.001). Lastly, children who think that their dental appearance is bad scored higher C-OIPD-U than those children who think their dental appearance to be good and average (*p* < 0.001) (Table 3).

Table 4 showed the results of discriminate validity testing. The results showed that C-OIDP-U can discriminate between normal and carious dental status, and the difference reached significance (*p* < 0.001). Children with DMFT + dmft ≥ 1 reported a higher C-OIDP-U score (11.7, SD = 9.6) when compared to caries-free children. Similarly, a pattern was observed in other clinical variables as Gingiva status (*p* < 0.001) and malocclusion (*p* < 0.001) had higher C-OIDP scores than their counterparts.

Table 5 showed the results of the inter-item correlation matrix among eight items of C-OIDP-U. No negative correlation coefficients were seen, and it ranged from 0.15 (representing between smiling and social) to 0.51 (between eating and cleaning) (Table 5).

## 4. Discussion 

In this study, C-OIDP was translated into Urdu language and tested for its reliability and validity on Pakistani 11–14-year-old school-going children for the first time. An appropriate measure to assess OHRQOL in the Pakistani pediatrics population was needed since the original English version of the C-OIDP measure would have not been suitable to use in different populations and cultures. The translation and adaption of C-OIDP in Urdu were carried out according to established guidelines and methodology provided in the literature [14]. In this study, the Urdu version of C-OIDP shows good reliability and validity among 11–14-year-old-school children in Pakistan.

The internal consistency of C-OIDP measured using Cronbach’s alpha coefficient value in this study showed comparable results with other validating studies, such as in Spain (0.68), Hungry (0.73), Turkey (0.73), Peru (0.65) and Chile (0.72), but it was lower than those values reported in Israel (0.81), Malaysia (0.80) and India (0.81) [20,21,22,23,24]. The value of Cronbach’s alpha coefficient was above 0.5 thresholds but lower than the 0.7 thresholds [25]. One of the factors that influence the value of alpha is the number of items in the scale, and fewer items in a measure usually provide a lower level of alpha, and if the items are added, the alpha value will increase [25]. The C-OIDP is a small measure having only eight items, which would have been the reason for the lower Cronbach’s alpha coefficient value in many studies. Although by increasing the items number of an instrument, alpha can be increased; however, brief, simple user-friendly and cost-effective Child OHRQoL instruments that can assess all dimensions would be more practical and acceptable for assessing OHRQoL in children [26].

In this study, the prevalence of impact on daily life was reported at 77.3% with at least one daily activity of children being affected. Previous studies reported mixed results with a high prevalence of oral impact on daily performances in Turkey (93.7%), India (93.8%) and Brazil (80.7%) and low prevalence in the UK (40.4%), (28.6%) and Indonesia (64.9%) [10,12,27,28,29].

Eating (40.3%) and cleaning (38.7%) were the two most prevalent oral impacted daily performance in this study. Eating is reported as the most prevalent impact by children in many other studies conducted on children in Turkey, Brazil, India, Peru and Croatia [12,21,30]. However, children in Malaysia and Indonesia reported cleaning teeth as the most impacted daily performance, which can be due to clinical oral health problems such as gingivitis, plaque accumulation and malocclusion [22,27]. This relationship of clinical oral problems with teeth cleaning was also reported in Sudan where children who reported oral impact on cleaning perceived impairments on gingival conditions such as bleeding and swelling [9]. In this study, 62.9% of the children had either unhealthy gingiva with redness or swelling and bleeding on probing, which can be a plausible reason for the high prevalence of impact on cleaning in this study. The results of prevalence of impact on daily performance also showed that social contact and smiling were among the least reported impact. Most children in developing countries are unaware of treatment options and opportunities for improvement of appearance due to a lack of oral health literacy and accessible dental care services [9].

The Urdu version of C-OIDP showed good construct validity in this study. It was able to discriminate levels of self-rated oral and general health, need for dental treatment and dental appearance and showed that if the perception of children about their oral health is better, the prevalence of oral impacts will be lower. This consistent pattern indicates that self-perceptions of oral health and its treatment needs along with dental appearance satisfaction are strongly associated with OHRQoL. These findings are similar to the studies in other countries where the self-perceived subjective measure of oral health showed a variation in OHRQoL similar to clinical normative measures [12,22,24].

There are some limitations and strengths of this study. The cross-sectional study design does not allow the verification of variables’ causes–effect relationships and any variation over time. The recruitment of children from only two public schools near the hospital and convenience sampling in this study can lead to bias and did not allow the generalization of results to the entire culturally diverse Pakistani population. Further longitudinal studies using a random sampling approach in recruiting the children from representative board-based populations are required to better understand OHRQoL measures in Pakistani children. Despite these limitations, this is the first time C-OIDP widely used OHRQoL instrument was translated into Urdu language and checked for its validity and reliability on Pakistani children. The questionnaire was administrated by conducting the face-to-face interview, which is considered a viable and reliable mode as compared to self-administration by younger children.

## 5. Conclusions

This study showed that the Urdu version of C-OIDP is a valid, reliable and efficient instrument of OHRQoL for use in the Pakistani population. This measure can provide valuable information to oral health care providers to identify children with poor oral health and its impact on their daily performance, which help policymakers in planning school-based oral health programs based on their needs and demands.

## Figures and Tables

**Table 1 children-09-00631-t001:** Demographic characteristics and prevalence of oral impacts on daily performances (Child-OIDP) in Pakistani children (*n* = 197).

Demographic Characteristics	*n* (%)	
**Age**		
**11**	39 (19.7)	
**12**	51 (25.8)	
**13**	65 (32.9)	
**14**	42 (21.6)	
**Gender**		
Male	115 (58.3)	
Female	82 (41.7)	
**Performance**	**Percentage of children with impact on performance** **(*n* = 197)**	**Mean Child-OIDP** **(±SD) on each performance (0 to 100)**
Eating	40.3	23.3 (26.3)
Speaking	17.5	7.3 (15.1)
Cleaning	38.7	15.6 (21.5)
Sleeping	16.2	5.3 (10.4)
Emotional Status	28.8	12.6 (25.2)
Smiling	12.6	8.3 (16.2)
Studying	10.2	3.7 (14.1)
Contact with people	9.1	6.1 (17.7)
At least one of the above	77.3	
Mean C-OIDP-U score		10.2 (8.1)

**Table 2 children-09-00631-t002:** Cronbach’s alpha, item-total correlation and alpha with deleted items.

Performance	Corrected Item-Total Correlation	Cronbach’s Alpha If Item Deleted
Eating	0.58	0.68
Speaking	0.43	0.65
Cleaning	0.43	0.64
Sleeping	0.51	0.67
Emotional Status	0.40	0.69
Smiling	0.42	0.71
Studying	0.37	0.70
Contact with people	0.35	0.72
Standardized Cronbach’s alpha = 0.69 Weighted kappa = 0.81		

**Table 3 children-09-00631-t003:** Findings for the construct validity of the C–OIDP-U.

Variables (Categories)	*n* (%)	Mean (SD)	*p*-Value
**Self-rated oral health**			<0.001
Good	104 (52.7)	5.3 (8.5)	
Fair	67 (34.2)	4.2 (6.3)	
Poor	26 (13.1)	7.5 (12.3)	
**Self-rated general health**			<0.01
Good	141 (71.5)	6.6 (11.3)	
Fair	36 (18.2)	5.8 (8.1)	
Poor	20 (10.1)	8.3 (9.5)	
**Self-rated need for dental treatment**			<0.001
No	106 (53.8)	7.4 (8.9)	
Yes	24 (12.1)	10.2 (8.9)	
Do not know	67 (34.0)	8.8 (7.8)	
**Dental appearance**			<0.001
Good	140 (71.1)	7.7 (10.3)	
Average	42 (21.3)	5.1 (7.8)	
Bad	15 (7.6)	9.3 (9.1)	

**Table 4 children-09-00631-t004:** Findings for discriminant validity of C–OIDP-U.

Variables (Categories)	*n* (%)	Mean (SD)	*p*-Value
**Dental status**			<0.001
DMFT + dmft = 0	36 (18.2)	5.6 (8.4)	
DMFT + dmft ≥ 1	161 (81.8)	11.7 (9.6)	
**Gingiva status**			<0.001
GI ≤ 1	73 (37.1)	12.6 (8.8)	
GI > 1	124 (62.9)	15.2 (7.5)	
**Malocclusion**			<0.001
Present	27 (13.7)	12.1 (8.6)	
Not Present	170 (86.2)	6.7 (9.1)	

**Table 5 children-09-00631-t005:** Findings for inter-item correlation matrix for reliability analysis.

	Eating	Speaking	Cleaning	Sleeping	Emotional Status	Smiling	Studying	Social
Eating	1							
Speaking	0.33	1						
Cleaning	0.51	0.25	1					
Sleeping	0.48	0.33	0.26	1				
Emotional Status	0.32	0.25	0.30	0.21	1			
Smiling	0.31	0.29	0.24	0.23	0.19	1		
Studying	0.22	0.20	0.21	0.25	0.21	0.18	1	
Social	0.25	0.19	0.18	0.17	0.16	0.15	0.20	1

## Data Availability

The data presented in this study are available upon request from the corresponding author.

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
