# Peer review of "Validation and Reliability Testing of the Child Oral Impacts on Daily Performances (C-OIDP): Cross-Cultural Adaptation and Psychometric Properties in Pakistani School-Going Children"

_children, 2022, doi:10.3390/children9050631_

Round 1

Reviewer 1 Report

Thank you very much for allowing me to review the original article entitled “Validation and reliability testing of the Child Oral Impacts on Daily Performances (C-OIDP): cross-cultural adaptation and psychometric properties in Pakistani school-going children” (children-1647758) . This manuscript is submitted for publication in the Section “Pediatric Dentistry”, in the Special Issue “Oral Health Related Quality of Life of Children”.

This is an oral health-related quality of life validation study of the Child-Oral Impact on Daily Performance (C-OIDP) to develop an Urdu version questionnaire for children 11-14 years ols) in Pakistan.

The title is informative and reflects the content of the article.

The aim of this study is to develop an Urdu version of Child-Oral Impact on Daily Performance (C-OIDP) and assess its reliability and validity for children's OHRQoL assessment in Pakistan.

Introduction:

The introduction uses adequate bibliography but could be supplemented with information on the known situation of oral health status of children in Pakistan.

The objective is well formulated, although I think it should be specified in children between 11 and 14 years of age.

Materials and Methods:

The approval of the ethics committee is presented.

It is presented as a cross-sectional design. Although data is collected from 6 months ago, I suggest that it is a cross-sectional and retrospective design.

I suggest presenting a flowchart showing the people who have been offered to participate, the participation rate and finally those who have accepted.

The validation has been carried out in a pilot study of 10 children. What criteria have selected only 10 children? This is somewhat strange since 197 children are available.

Results:

In Table 1, a test should be carried out to indicate whether there are significant differences by age, gender, and performance.

Discussion: It is well described, and the conclusions are in line with the results of the work.

Author Response

Dear Editor and Reviewers,

We would like to thank you for the valuable comments and suggestions on the report “Validation and reliability testing of the Child Oral Impacts on Daily Performances (C-OIDP): cross-cultural adaptation and psychometric properties in Pakistani school-going children".  We believed that all these had positively contributed towards improving the quality of the report itself and hope that these will help in getting the report to be published in the Children journal.

I am resubmitting our revised manuscript after addressing point-by-point, the comments and suggestions from the reviewers. We also would like to thank you in advance for agreeing to review the resubmitted version of the manuscript.

Thank you.

Dr. Farooq Ahmad Chaudhary

School of Dentistry, SZABMU.

Reviewer-1

Comment 1: Thank you very much for allowing me to review the original article entitled “Validation and reliability testing of the Child Oral Impacts on Daily Performances (C-OIDP): cross-cultural adaptation and psychometric properties in Pakistani school-going children” (children-1647758) . This manuscript is submitted for publication in the Section “Pediatric Dentistry”, in the Special Issue “Oral Health Related Quality of Life of Children”. This is an oral health-related quality of life validation study of the Child-Oral Impact on Daily Performance (C-OIDP) to develop an Urdu version questionnaire for children 11-14 years ols) in Pakistan.

Response: Noted and thankyou.

Comment 2: The title is informative and reflects the content of the article. The aim of this study is to develop an Urdu version of Child-Oral Impact on Daily Performance (C-OIDP) and assess its reliability and validity for children's OHRQoL assessment in Pakistan.

Response: Noted and thank you.

Comment 3: Introduction: The introduction uses adequate bibliography but could be supplemented with information on the known situation of oral health status of children in Pakistan.

Response: To best of our knowledge there are no studies conducted on OHRQoL of children in Pakistan and in introduction section the background on OHRQoL and instruments used for measuring OHRQoL were discussed. No OHRQOL measure for children was available in the Urdu language before this study which we can cite and discus in this section. Revised Oral health status of children in various countries were discussed in the discussion section.

Comment 4: The objective is well formulated, although I think it should be specified in children between 11 and 14 years of age.

Response: Revised and improved as suggested (line 67).

Comment 5: Materials and Methods: The approval of the ethics committee is presented.It is presented as a cross-sectional design. Although data is collected from 6 months ago, I suggest that it is a cross-sectional and retrospective design.

Response: Revised and improved as suggested (line 69).

Comment 6: I suggest presenting a flowchart showing the people who have been offered to participate, the participation rate and finally those who have accepted.

Response: This study used opportunity sampling technique to recruit  197 children aged 11–14-year-old.  We have recruited anyone who is available and willing to take part in the study, therefore there are no dropouts or left outs. We approach 197 children and the data of all 197 children were used in this study. Therefore, flowchart cannot be made for this type of sampling and recruitment.  

Comment 7: The validation has been carried out in a pilot study of 10 children. What criteria have selected only 10 children? This is somewhat strange since 197 children are available.

Response: In the published literature there is no hard and fast rule to decide the number of samples to take in your pilot study. Some recommend obtaining approximately 10 participants (Nieswiadomy, 2002) or 5-10% of the final study size (Lackey & Wingate, 1998). Similar recommendations can be found in texts in other areas of clinical and epidemiology research (Hulley et al., 2001).

The final decision to be guided by cost and time constraints as well as by size and variability of the population. Reference of these recommendations is given below.

  1. Hertzog, M.A., 2008. Considerations in determining sample size for pilot studies. Research in nursing & health31(2), pp.180-191.
  2. Nieswiadomy, R.M. (2002). Foundations of nursing research (4th ed.). Upper Saddle River, NJ: Pearson Education.
  3. Hulley, S.B., Cummings, T.B., Browner, W.S., Cummings, S.R., Hulley, D.G., & Hearst, N. (2001). Designing clinical research: An epidemiological approach. Philadelphia: Lippincott, Williams, & Wilkins
  4. Lackey, N.R., & Wingate, A.L. (1998). The pilot study: One key to research success. In P.J. Brink & M.J. Wood (Eds.), Advanced design in nursing research (2nd ed.). Thousand Oaks, CA: Sage.

Comment 8: Results: In Table 1, a test should be carried out to indicate whether there are significant differences by age, gender, and performance.

Response: The objective of the study was to develop an Urdu version of Child-Oral Impact on Daily Performance (C-OIDP) and assess its reliability and validity for children’s OHRQoL assessment in Pakistani. All the required tests and analysis (Cronbach's alpha, item-total, Pearson correlation coefficients, weighted kappa coefficient, construct and discriminant validity using ANOVA and independent t-test) were performed to assess its reliability and validity. The age and gender difference can be seen in Table 1 by the percentages and numbers. Specific test to see any difference for these two variables is unnecessary as its as not the aim of this study.

Comment: Discussion: It is well described, and the conclusions are in line with the results of the work.

Response: Noted and thank you.

Reviewer 2 Report

The investigation conducted by the authors is well executed and follows a correct scientific methodology. The cross-sectional study has been correctly structured and the results obtained are interesting. The importance of the relationship between oral health and quality of life in the child population is significant and must be carefully investigated. The tables are clear and easy to read, I only advise the authors to graphically improve table 1, the writings are not well aligned (Male-Female). I also suggest to the authors to implement the bibliography, I recommend reading the following articles, which have also carried out surveys on a child population: "Gandini P, et al. Statistical survey of malocclusion in school age children. Mondo Ortod. 1989 Jan -Feb; 14 (1): 73-8. "; "Capocasale G, et al. How to deal with coronavirus disease 2019: A comprehensive narrative review about oral involvement of the disease.Clin Exp Dent Res. 2021 Feb; 7 (1): 101-108." and "Lucchese A, et.Wiskott-Aldrich syndrome: Oral findings and microbiota in children and review of the literature.Clin Exp Dent Res. 2022 Feb; 8 (1): 28-36."

Author Response

Dear Editor and Reviewers,

We would like to thank you for the valuable comments and suggestions on the report “Validation and reliability testing of the Child Oral Impacts on Daily Performances (C-OIDP): cross-cultural adaptation and psychometric properties in Pakistani school-going children".  We believed that all these had positively contributed towards improving the quality of the report itself and hope that these will help in getting the report to be published in the Children journal.

I am resubmitting our revised manuscript after addressing point-by-point, the comments and suggestions from the reviewers. We also would like to thank you in advance for agreeing to review the resubmitted version of the manuscript.

Thank you.

Dr. Farooq Ahmad Chaudhary

School of Dentistry, SZABMU.

Reviewer-2

Comment: The investigation conducted by the authors is well executed and follows a correct scientific methodology. The cross-sectional study has been correctly structured and the results obtained are interesting. The importance of the relationship between oral health and quality of life in the child population is significant and must be carefully investigated.

Response: Noted and thankyou

Comment: The tables are clear and easy to read, I only advise the authors to graphically improve table 1, the writings are not well aligned (Male-Female).

Response: Noted and corrected as suggested.

Comment: I also suggest to the authors to implement the bibliography, I recommend reading the following articles, which have also carried out surveys on a child population: "Gandini P, et al. Statistical survey of malocclusion in school age children. Mondo Ortod. 1989 Jan -Feb; 14 (1): 73-8. "; "Capocasale G, et al. How to deal with coronavirus disease 2019: A comprehensive narrative review about oral involvement of the disease.Clin Exp Dent Res. 2021 Feb; 7 (1): 101-108." and "Lucchese A, et.Wiskott-Aldrich syndrome: Oral findings and microbiota in children and review of the literature.Clin Exp Dent Res. 2022 Feb; 8 (1): 28-36."

Response: The above mentioned first reference Gandini P, et al. was done in 1989, that is too old to be cited also not related to this study. The second recommended reference Capocasale G, et al. 2021 is related to coronavirus disease and not related to our current study, and lastly reference is also not related to current study. Therefore, these references cannot be cited in this study as they don’t have any relevance to our study.
